# Perceptions of Resettled Refugee Congolese Women: Maintaining Cultural Traditions during Resettlement

**DOI:** 10.3390/ijerph192416714

**Published:** 2022-12-13

**Authors:** Chelsey Kirkland, Na’Tasha Evans, Kamesha Spates, Cedric Mubikayi Kabasele

**Affiliations:** 1Center for Public Health Systems, University of Minnesota School of Public Health, 420 Delaware St SE, Minneapolis, MN 55455, USA; 2College of Public Health, Kent State University, 800 Hilltop Dr., Kent, OH 44242, USA; 3School of Health Sciences, College of Education, Health, and Human Services, Kent State University, 1225 Theatre Dr., Kent, OH 44242, USA; 4Department of Sociology and Criminology, Kent State University, 700 Hilltop Dr., Kent, OH 44242, USA; 5Department of Africana Studies, University of Pittsburgh, 230 S. Bouquet St., Pittsburgh, PA 15260, USA

**Keywords:** Congolese, women, refugee, acculturation, qualitative

## Abstract

Conflict-displaced refugees have increased significantly globally. The Democratic Republic of Congo is the leading country with refugees in the United States, where many resettle in Ohio. Women refugees are highly vulnerable, yet little literature has focused on them. Furthermore, maintaining cultural traditions can provide comfort during the tumultuous resettlement process. Therefore, this study used mixed methods to understand the perceptions of Congolese refugee women on maintaining cultural traditions during resettlement in Ohio. Translator-assisted, orally administered demographic survey and face-to-face interviews were conducted among resettled Congolese refugee women (*n* = 20) 18 and older, who arrived in the United States from 2011 to 2018, and were currently receiving Ohio resettlement agency assistance. Researchers applied descriptive coding and thematic analysis to identify themes and subthemes. Three themes were identified among the resettled Congolese refugee women regarding maintaining cultural traditions in the United States. The three themes comprised (1) clothing and dressing, (2) food, and (3) parenting style. Our work examined resettled refugee Congolese women’s perceptions of maintaining their culture after resettling in Ohio. These study findings could assist community engagers with insights and practical recommendations on supportive services for resettled Congolese women and a deeper understanding of complex acculturative situations facing them during resettlement.

## 1. Introduction

According to the United Nations High Commissioner for Refugees (UNHCR), refugees are persons who have fled war, violence, conflict, or persecution; they have crossed an international border for safety in another country often leaving behind homes, possessions, jobs, and family and friends [1]. By the end of 2020, 20.7 million people were considered refugees [1]. One of the highest refugee-producing countries is the Democratic Republic of Congo due to ongoing armed conflict and political instability [2].

This armed conflict within the Democratic Republic of Congo resulted in continued internal fighting throughout the country since winning its independence in 1960 [3]. Despite the signed Lusaka Ceasefire Agreement, a peace deal signed in 2002 [4], and the formation of a transitional government in 2003, violence by armed groups has continued against civilians in the eastern region of the Democratic Republic of Congo [5]. This is partly due to poor governance, weak institutions, and rampant corruption [5]. In addition, conflict continues to mount over an estimated USD 24 trillion worth of untapped mineral resources [5], resulting in nearly 5 million people being displaced between 2009 and 2019 [3].

The Department of Homeland Security tracks refugees coming to the United States, showing where refugees arrived from, which states the refugees initially resettled in, and the demographic characteristics of the refugees. For example, the United States increased its refugee admission from 15,000 to 62,500 for Fiscal Year 2021, with 22,000 being allocated from the continent of Africa. In Fiscal Year 2020, the United States resettled almost 12,000 refugees, with about 3000 being from the Democratic Republic of Congo, the largest number of refugees resettled from any country [6].

A closer look at where refugees are received and then resettle in the United States reveals that Ohio has some of the highest numbers of Congolese (1392 received since FY 2019) [6,7,8]. Most of the refugees received in Ohio in the past five years have been from the Democratic Republic of Congo, even during the COVID-19 pandemic [9]. Yet, despite Ohio’s growing refugee population, little research exists about the refugees that have resettled. One of the few existing studies was conducted by Dutt and colleagues. They surveyed 280 refugees to compare the experiences of African refugees (Burundian, Congolese, and Somali) to the experiences of Bhutanese, Latinx, and Middle Eastern on several aspects of civic life in Cincinnati, Ohio [10]. Findings suggest that African refugees report more negative outcomes than other refugees. Findings also suggest that refugees in neighborhoods with a higher percentage of Black residents reported less satisfaction overall. An additional study by Mahoney et al. [11] examined the challenges of Congolese refugees in Akron, Ohio, and Tampa, Florida. They revealed when working with refugees from the Congo Wars and focused on the timing of arrival in the United States, community and class, schooling, gender and family, and food and diet. In particular, they found that unmarried mothers remained some of the most marginalized groups and women, who tended to have extremely limited education and literacy, were afraid to ask for help for fear of how they may be seen. Additionally, while traditional foods may be difficult to find in grocery stores, many refugees did not want to participate in community gardens due to wanting to leave that hard lifestyle behind, leading to a dependence on cheap American foods and lack of fresh vegetables in their diet.

Over 50 percent of the Congolese refugee population are women and children. These two groups are considered the most vulnerable [3]. In fact, approximately 20 percent of Congolese refugee women in the United States are categorized as “women-at risk” [12]. This categorization is given to women or girls lacking physical protection, most often provided by a male family member [13]. In addition, migration frequently disrupts their highly valued social circles, and loss of extended communities and social isolation due to immigration may create mental health challenges [12,14,15]. Women also fare worse on most indicators with inequalities related to language acquisition, health, and employment, enduring or becoming worse over time due to gender stereotyping and discrimination within the community and policymakers [16]. Furthermore, the needs of refugee men and women differ significantly [16].

In particular, Congolese refugee women come from historically patrilineal, patrilocal, and patriarchal cultures, in which man dominance controls family decisions and finances [11]. When couples encounter American gender relationships, such as women learning about legal protections and the ability to divorce, discord can occur, and tensions tend to increase with time spent in the United States [11]. Single Congolese refugee women encounter even more challenges. These challenges are especially burdensome for unmarried mothers in the Democratic Republic of Congo [11].

Despite these challenges, one avenue to improve the health and well-being of Congolese refugee women may lie in their acculturation process. Acculturation Theory is well established when explaining outcomes around contact with culturally dissimilar people, groups, and social influences [17,18]. Schwartz et al. [18] suggest that acculturation generally focuses on immigrants, refugees, and asylum seekers. This has evolved from being viewed as a unidimensional process where heritage retention was in opposition to the receiving culture to one encompassing four different categories of acculturation: (1) assimilation (adopt a new culture and discard heritage), (2) separation (reject new culture and retain heritage), (3) integration (adopt a new culture and retain heritage), and (4) marginalization (reject new culture and heritage) [18]. Research has shown that the type of acculturation experienced by refugees’ experiences can impact their well-being and ability to adapt to their new surroundings [19]. Literature suggests that the more an immigrant can acculturate in the way they desire, the better they report their well-being due to experiencing less acculturative stress [20]. However, there is a shortage of literature on Congolese refugee women acculturation. Therefore, we used Acculturation Theory within the study, and below, we examine the literature on Congolese traditions and current refugee support and mental health in the United States to frame the study context.

Upon arriving in the United States, refugees immediately face a completely different way of life, from language to the clothes worn, to the food eaten on top of the challenging circumstances from which they came [21]. When facing this different way of life, refugees begin to experience acculturation. When refugees experience integration acculturation they tend to adapt to their surrounding circumstances better [17] and experience better health outcomes and well-being due to experiencing less acculturation stress [19,22].

### 1.1. Congolese Traditions

Culture can be defined as behavior patterns arising from social learning in specific social relationships within specific social groups [23]. These behavior patterns can rapidly become frequent in a population and form cultural traditions [23]. Therefore, it is imperative for persons within a culture to adapt to macrocultures while keeping their heritages (i.e., cultural practices) vibrant, which can prevent them from being suppressed or extinguished [24]. There is a wide array of cultural traditions, including language, food, dressing, and family lifestyles. While French is the official language of the Democratic Republic of Congo, only about 47 percent of the population can read, write, or speak French [25]. The Democratic Republic of Congo has 400 tribes and over 242 other spoken languages, among which four national languages are Kikongo, Lingala, Tshiluba, and Swahili. In addition, most Congolese speak several languages depending on their education level and the region they are from or have traveled to.

Cassava and maize are the main staple foods, including the tuber, harvested, dried, and milled. The cassava and corn flour are then used to make a starchy paste or mash called fufu or ugali, used in almost every meal [25]. Depending on the region, there may be a slight difference in the preferred type of flour or the way to make fufu. These meals are usually accompanied by fish or meat stew, depending on wealth, season, and availability, with a side of vegetables such as cassava leaves, okra, spinach, or mushrooms [26]. Other common foods include rice, sweet potatoes, taro, yam, plantains, tomatoes, beans, and groundnuts [25].

Compared to the national average, refugees resettled in the United States have consistently reported low food security rates. Sub-Saharan African refugees, including Congolese refugees, have reported even lower rates than other groups of refugees [27,28,29]. This is due to dietary acculturation and limited access to safe and culturally appropriate food [30,31].

There are complexities when referring to a singular “Congolese” community due to the numerous languages spoken, which is observed in differing individuals’ and family’s definition of being a member of the ‘Congolese community’ [11]. Therefore, since communities often have different lifestyle traditions, a challenge is created when describing the customs of Congolese families. The tradition in this region and literature is that women are expected to marry, bear, and parent the children. At the same time, men are the head of the household [25], which presents additional burdens on Congolese women refugees.

Dressing traditions in the Republic Democratic of Congo have vastly changed over time. Before colonization and European influence, the Congo Kingdom was renowned for the extraordinary quality of raffia fabric or palm clothing they produced. However, cotton-based clothes became popular with the arrival of the Europeans [32].

Cotton-based, factory-printed textiles are trendy in many sub-Saharan African countries, including the Democratic Republic of Congo. There are two main types: wax prints and roller prints. Jointly they are called “page”, a French word meaning loincloth, referring to cloth worn as a wrapper [33]. They usually have vibrant colors and patterns and can be worn depending on the region. Like many Africans, Congolese people wear the loincloth to valorize their culture, identity, and origins. These pieces of clothing vary in value, depending on the quality of the material used and whether they are imported or produced locally. While the European influence is still strong, especially in urban areas and among the youth, there is a sense of pride that comes with wearing the loincloth [32]. Congolese have endured numerous challenges and essentially live a survival life [25]. Despite continued hardships, Congolese often wear their best clothes to greet guests and serve their best available food [25].

### 1.2. Refugee Support and Mental Health in the United States

Once refugees arrive in the United States, the local resettlement affiliate, family members, or friends take the individuals to their initial house with essential furnishing, appropriate food, and other necessities [34]. While refugees do receive assistance for initial resettlement (such as employment services, applying for social security cards, and connecting with social and language services), this support is only provided by the government’s Reception and Placement Program for the first three months after arrival and are dependent on other non-governmental organizations for long-term assistance [34]. Despite the high prevalence of refugees having experienced traumatic experiences during their pre-immigration and acculturation stress during resettlement, these support services do not extend to assisting the mental health of refugees [35], despite United States resettlement potentially causing additional stressors and living difficulties [36].

Mental health is of critical concern for refugees [37,38], and the Centers for Disease Control and Prevention [39] lists mental health as one of the priority health conditions for Congolese refugees. This is because many refugees are exposed to traumatic events such as displacement, torture, murder, incarceration, loss of family members, sexual assault or exploitation, starvation, disease, and lack of shelter [35]. Refugees resettled in Western countries may have symptoms of post-traumatic stress disorder about ten times greater than the general population [35]. These challenges continue during their resettlement process and can be exacerbated due to acculturative difficulties (e.g., cultural and language barriers, stigma, lack of resources and information) and limited coping mechanisms upon resettlement [35,40]. Dutt et al. [10] found that African refugees within Cincinnati, Ohio, report experiencing more challenges and dissatisfaction than non-African refugees. One clear avenue to assist with these mental health challenges is assisting refugees with maintaining aspects of their culture of origin, such as traditional cultural practices, while incorporating their host society’s culture (i.e., acculturation integration). One’s ability to maintain culture is associated with enhanced well-being in refugees and immigrants [19]. Despite this clear association, there is a dearth of literature on the acculturation of Congolese refugee women, the highest percentage of refugees coming to the United States, and some of the most vulnerable [3,6].

### 1.3. Current Study

Congolese refugee women’s resettlement within the United States is a significantly under-researched area, and literature on cultural integration experiences is also limited [41]. Therefore, this study aims to provide Congolese refugee women the opportunity to describe which cultural traditions they have been able to maintain once resettled in Ohio. Thus, our research question was: what cultural pieces have newly resettled Congolese women kept after resettling in the United States? Therefore, this study’s unique contribution is adding Congolese refugee women’s experiences [16] to Acculturation Theory literature and being one of the first to explore which cultural traditions they have been able to maintain after resettling in Ohio.

## 2. Materials and Methods

### 2.1. Study Design and Participants

The overall study design was convergent mixed methods where the quantitative and qualitative strands were collected simultaneously [42]. We used descriptive or interpretive qualitative methods [43,44], collected via semi-structured individual interviews. Quantitative data were collected through a demographic survey, which was orally administered by a translator due to language barriers among participants. Additionally, we used Acculturation Theory to frame the interview questions and obtain a holistic understanding of which cultural traditions have been maintained by Congolese refugee women while resettling in Ohio. Although Acculturation Theory has been researched extensively, little research has examined it within Congolese refugees.

We worked with a resettlement agency whose employees recruited participants by distributing study-related information to target participants when they came to the agency for services. The resettlement agency provided a translator, interviewing space, and facilitated interview scheduling. Twenty women participated in this study. Five factors guided eligibility: (1) self-identifying as a Congolese refugee, (2) self-identifying as a woman, (3) being 18 years of age or older, (4) residing in Ohio, and (5) having arrived in the United States in 2010 or later.

### 2.2. Data Collection

Our study was approved by the Institutional Review Board at the authors’ university (IRB #18-198). Interested participants were referred to the agency translator at our partner resettlement agency, who screened them for eligibility. All eligible participants signed up for existing interview time slots.

Interviews occurred during the summer of 2018 in a private room at a resettlement agency with a woman translator from the agency who was fluent in the participants’ native language, which provided a trusting environment for the interviewees. The translator obtained verbal informed consent from each participant before starting each interview. Interviews began with the orally administered demographic survey to understand participants’ backgrounds and then followed the semi-structured interview guide. Semi-structured interviews are flexible and were beneficial for this study, as they allowed the interviewers to gather contextual information as needed through prompts and follow-up questions [45,46,47,48,49]. All interviews were audio recorded and conducted by a research team member with the translator’s assistance.

At the beginning of each interview, the explanation and study purpose were explained to the participant, who was provided the opportunity to ask questions about the study or procedures. The participant was also reminded that the interview would be audio recorded for transcription and that their answers would not be shared with anyone else, including their healthcare provider or agency, and it would not affect their current or future care. Each interview lasted about 60 min, and participants received a USD 10 cash incentive after their interview.

### 2.3. Measures and Analysis

All participants verbally completed the demographic questionnaire at the start of their interview. This questionnaire assessed age, employment status, whether they spent time in a refugee camp(s), number of refugee camps, length of time in a refugee camp(s), location of refugee camp(s), arrival to the United States, and who came with them to the United States. The research team developed the semi-structured interview guide based on a literature review.

Demographic questionnaires were analyzed in SPSS (IBM Corp. Released 2017. IBM SPSS Statistics for Windows, Version 25.0. Armonk, NY, USA: IBM Corp.) using descriptive statistics. All interview audio files were transcribed verbatim and applied inductive descriptive and thematic analysis techniques [49,50] by authors C.K. and C.M., then reviewed by author NE using QSR International’s NVivo 12 qualitative data analysis software. The first round of coding involved C.K. and C.M. developing a descriptive code. This technique assigns labels to data to summarize it in a word or short phrase on the basic topic for each meaning unit, as described by Saldaña [49]. Subsequent analysis rounds involved C.K. and C.M. using descriptive codes to develop overarching themes [49]. Both descriptive coding and thematic analysis were deemed appropriate for this research based on Saldaña’s [49] recommendation to use it in exploring participants’ psychological world of beliefs, constructs, and emotional experiences. All authors used intercoder reliability to confirm themes.

### 2.4. Research Team Positionality Statement

The first author holds a doctorate in Public Health, specializes in social determinants of health, and identifies as a white woman. The second and third authors are United-States-born Black women who hold doctorate degrees and have extensive research and background examining health disparities and minority health. The fourth author is a native of the Democratic Republic of Congo who holds a Master’s in Public Health, is a doctoral student, and has a background in health disparities and epidemiology.

## 3. Results

### 3.1. Sample Characteristics

We interviewed 20 participants ranging in age from 19 to 68 years of age. The mean was 45 years, with a standard deviation of 13.8 years. Most participants were unemployed and spent time in one refugee camp, the most common camp was Rwanda, and the time they resided there ranged from one to 22 years. Lastly, most participants arrived in 2016 in the United States with one or more family members. Please see Table 1 for all participant demographic characteristics.

### 3.2. Qualitative Results

Our findings demonstrate that the traditions Congolese refugee women have maintained since resettling in Ohio were (1) clothing and dressing, (2) food, and (3) parenting style. These themes, definitions, frequency, and supporting sample quotes are in Table 2. Participants shared a wide array of maintained cultural traditions; therefore, only the most salient themes are discussed.

#### 3.2.1. Theme 1: Maintaining Clothing and Dressing

Thirty percent (*n* = 6) of participants’ responses to maintaining cultural traditions were related to clothing and dressing. Interview data coded in this category indicated words or phrases that highlighted participants’ current clothing or dressing choices compared to living in Africa. For instance, a 55-year-old described her clothing choices, saying, “…*yeah, we dress in traditional garb like gowns. Some of them I came with from Africa, I still wear them, and there is no one who stops me to ask me, hey how are you dressed? And that is good. I can get pants but I can’t ever wear them*.” In a similar way, another 35-year-old woman shared, “…*in dressing, I came here and I don’t wear pants you can see my African dresses*.” Likewise, a 42-year-old stated, “*I have continued to wear my African clothing or African attire, and people usually bring a lot of, um, traditional clothes, or like African wear from back home*.” Through these examples, it was clear that maintaining clothing and dressing culture traditions provided participants with a foundation for their acculturation process.

#### 3.2.2. Theme 2: Maintaining Cultural Foods

Twenty percent (*n* = 5) of participants discussed maintaining their food traditions after arriving in the United States. Interview data in this category included words or phrases that highlighted participants’ current food choices compared to living in Africa. One 35-year-old woman stated, “…*I can’t eat the American food, I cook my own food such as yams, greens, beans and rice*.” Another 47-year-old woman shared, “*The food we ate in Africa is cabbage, fish, ugali, beans which we can get here. In Africa I was a livestock farmer and was able to find anything I needed. But I’m able to find some of the foods here*.” Similar to the first theme around clothing and dressing, maintaining cultural foods was another basis for the participants’ acculturation process.

#### 3.2.3. Theme 3: Maintaining Parenting Style

Twenty percent (*n* = 5) of participants’ responses about keeping their cultural traditions were related to parenting style. Interview data coded in this category indicated words or phrases that highlighted participants’ parenting style compared to living in Africa. For instance, a 36-year-old stated that she has continued “…*taking care of my children well, and another one is telling them and showing them the importance of education and continuing to follow the laws of this country so we can live better than we did back home*.” Lastly, a 48-year-old described the differences between her parenting style maintained from Africa and those in the United States by saying, “*You know, Africa is different from here. Because the first one is child rearing. We came here to find that you can’t [discipline] your child if they make mistakes that culture for us it is a no, because in our culture, even if it is a young man like this one makes a mistake, we [discipline] him, even if he is married and does something that displease the parents be beat him because, you are never older than your parents*.” Parenting is challenging for all persons but can be particularly hard for newly resettled refugees trying to understand the new location’s social norms. Merging their previous parenting styles with those expected in the new location may be a particular barrier along their acculturation journey.

## 4. Discussion

In this study, we used Acculturation Theory to provide Congolese refugee women the opportunity to describe which cultural traditions they have been able to maintain once resettled in Ohio. Most Acculturation Theory literature focuses on immigrants, refugees, and asylum seekers, whereas we expand this theory’s use to incorporate Congolese refugee women’s maintained traditions in the United States. In addition, this theory provided us with a framework for our interview guide and obtained a holistic understanding of their experiences.

Some of our results are consistent with the literature, while others expand upon it. Findings demonstrate that Congolese refugee women have maintained cultural traditions while resettling in Ohio are based on three overarching themes: (1) clothing and dressing, (2) food, and (3) parenting style.

Most participants described retaining their Congolese clothing or dressing styles when examining clothing and dressing. This finding expands the literature. Pavlish [51] found that clean and colorful clothing was essential to Congolese women and that their Kitenges seemed to contribute positively toward their identity as women and provide a social status. We recommend agencies working with resettled Congolese refugee women to understand the importance of maintaining their clothing. This could include helping them repair their current dresses or purchasing new ones. Furthermore, this assistance is one avenue to facilitate cultural integration [18].

Most participants described retaining their traditions around the types of food they consumed in the Democratic Republic of Congo. This finding is consistent with the literature. Dietary acculturation is the transition during which refugees adopt dietary habits (e.g., food types, consumption patterns, preparation patterns) of the new county [30]. Refugees report dietary acculturation challenges. These challenges were often due to limited access to traditional foods, shopping practices, language barriers, and economic resources to purchase food [30]. McElrone et al. [30] elaborate that many Congolese and Burundian refugees desire to continue consuming their traditional foods and will go to great lengths to access them. One of the few studies in Ohio also found that Congolese cherish their traditional foods [11]. From an integration Acculturation Theory perspective, and also recommended by McElrone et al. [30], we suggest organizations provide Congolese refugees with orientation services to local grocery stores that would offer them opportunities to find local food similar to their traditional foods. We also recommend adding culturally appropriate foods at local grocery stores to aid dietary acculturation. Some groups have begun community gardens for refugees; however, few Congolese refugees have lived on farms since the 1990s, and many are happy to leave that behind and, thus, look down on farming [11].

Our final theme revolved around parenting, in which participants described being able to retain the same parenting styles as in Africa. This finding is consistent with literature showing that while parenthood is a significant challenge for refugee parents, often due to previous and current traumatic experiences compounded with challenges local parents face, acculturation experiences add additional stress [52]. Therefore, when refugees experience acculturation integration and receive parenting support, their adjustment to the new country is significantly improved [52], potentially benefiting their mental health as well. Our participants shared similar sentiments around the strong desire to retain their parenting style while acclimating to the new local culture. Previous acculturation research also found that family life, which encompasses parenting, is strongly related to one’s attitude towards sociocultural maintenance and that family relationships are particularly important for continuation of ethnic culture [53]. Additionally, family unity and cohesion are important indicators of individual mental and relational health within collectivistic cultures, such as the Congolese culture, and fostering family connection and interdependence through supporting refugee parenting styles during acclimation is paramount [54]. Therefore, we recommend that refugee resettlement agencies follow Ochocka and Janzen’s [55] framework when working with resettled Congolese refugee women’s parents. This framework assists one in identifying relevant variables related to immigrant parenting and consists of cultural parenting orientation, parenting styles, host country context, modifications of orientation and styles, parenting contribution, and parenting support.

### 4.1. Contributions to the Literature

Our study has four primary contributions to the literature by (1) being one of the first to exclusively focus on describing cultural traditions maintained among Congolese refugee women in Ohio, a prominent destination, (2) focusing exclusively on Congolese refugee women, and (3) incorporating Acculturation Theory into this understudied topic. For example, while Moinolmolki et al. [52] focused on Congolese refugees in the Mid-Atlantic United States, they included Somali and Bhutanese refugees and only examined parenting challenges. Furthermore, when Pavlish [51] explored the meaningful life experiences of Congolese and found the importance of maintaining clothing traditions, it included men, it was conducted with Congolese refugees residing in a Rwandan refugee camp and did not have salient themes around food traditions. Lastly, as Joyce and Liamputtong [22] and Phillimore [21] corroborate, little attention has been paid to the varied settlement experiences of individual refugees and acculturation, particularly among Congolese refugee women.

### 4.2. Limitations

We acknowledge a few study limitations. First, our study had a small sample size in which participants’ backgrounds varied greatly, limiting the generalizability of our findings. However, by using qualitative methods, we gained a deeper understanding of the traditions maintained by participants. Secondly, all participants resided in Ohio, which may influence which traditions participants could maintain. For example, Ohio has many culturally appropriate grocery stores, allowing Congolese refugee women to maintain food traditions that may not be available in other states. Therefore, we recommend that future research examine maintained traditions among Congolese refugee women in other states and contexts as they may differ. We also recommend future research comparing preserved cultural traditions among different groups of refugees to understand how needs differ. In addition to future research focusing on acculturation and Congolese refugee women, research may explore the interaction between place attachment and acculturation and their effects on this population. Lastly, research should explore the intersection of cultural competence and acculturation, particularly within social, political, and economic contexts [56].

## 5. Conclusions

The three themes we identified among resettled Congolese refugee women regarding maintaining cultural traditions in the United States were (1) clothing and dressing, (2) food, and (3) parenting style. Overall, our study findings provide insights and practical recommendations for practitioners and researchers who offer supportive services to resettled Congolese women. These results also provide a deeper understanding of complex acculturative situations Congolese women experience during resettlement.

## Figures and Tables

**Table 1 ijerph-19-16714-t001:** Demographic characteristics of participants (*n* = 20).

Characteristics	%	*n*
How long have you been in the United States?		
2011	5%	1
2015	10%	2
2016	45%	9
2017	5%	5
2018	3%	3
What is your employment status?		
Not working	90%	18
Full-time	10%	2
Did you spend time in a refugee camp?		
No	20%	4
Yes	80%	16
How many refugee camps were you in?		
0	20%	4
1	50%	10
2	25%	5
3	5%	1
Where were the camps located? Where did you go as a refugee?		
Uganda	25%	5
Rwanda	40%	8
Burundi	10%	2
Tanzanisa	20%	4
Multiple Locations	5%	1
Who did you come with?		
Self	10%	2
Child	5%	1
Multiple children	20%	4
Husband and child or children	40%	8
Sibling	10%	2
Sibling and child or children	5%	1
Parent(s) and child or children	5%	1
Parent(s), sibling(s), child or children	5%	1

**Table 2 ijerph-19-16714-t002:** Theme table.

Theme	Definition	Frequency of Theme *n* (%)	Sample Quote
Clothing and Dressing	Words or phrases that highlighted participants’ current clothing or dressing choices compared to living in Africa.	6 (30%)	“In dressing we wear robes/dresses. But I haven’t worn pants yet. We still wear the long cotton African dresses that we brought with us from Africa and some we have bought from here.” 42-year-old
Food	Words or phrases that highlighted participants’ current food consumption compared to living in Africa.	5 (20%)	“I still get the same food, hmm whenever I can food from back home-fish, hmm beans, any kind of vegetables that we would get back home. But there is a certain kind of sweet potato that we get back home, but here it tastes different, and I like the ones from back home.” 68-year-old
Parenting style	Words or phrases that highlighted participants’ parenting style compared to living in Africa.	5 (20%)	“One custom that I have continued to follow is, especially when it comes to disciplining my children, when they do something I always question them and give them advice on what’s important and what to follow.” 44-year-old

## Data Availability

The data are not available due to concerns around data confidentiality of participants.

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
