# Peer review of "Perceptions of Resettled Refugee Congolese Women: Maintaining Cultural Traditions during Resettlement"

_ijerph, 2022, doi:10.3390/ijerph192416714_

Round 1
Reviewer 1 Report
General comments
The authors provide a study using Acculturation Theory to examine the refugees maintaining cultural traditions. I have provided the following feedback I hope is found to be helpful. Overall, I have some improvements suggested for general layout, the need to proofread, the structure, and most importantly the study methods.
Line 34: full stop at beginning of sentence. By
Line 65: possibly missing the word working 'with'? Also, the sentence reads a bit incongruent.
These are just some examples; the paper needs a good proofread, at times it is difficult to follow the argument and some sentences are running on.
Abstract
Can the authors provide a structed abstract, purpose, methods, results, conclusion, or variation of this, as the abstract could be improved on to make findings and conclusion clearer. They may also wish to reconsider methods based on feedback
Introduction
The introduction section can probably be shortened a little, it seems to run on
P.3 Line 104: Might be better suited to the materials methods section, seems out of place when next paragraphs are still discussing theory as part of introduction, I would also place positionality statement here
1.1 to 1.3 can be discussed under the introduction heading, or possibly just have a 1.1 for Acculturation Theory and present under this heading
1.2 and 1.3 Mental Health are both discussing the same thing, merging all these heading and cutting down on word count might help. With regards to prevalence of MH with refugees the authors are missing several important studies, for example, you may wish to cite one or some of the following
1.4 study context will thus need to be changed to 1.1 or 1.2 depending
Henkelmann, J.R., de Best, S., Deckers, C., Jensen, K., Shahab, M., Elzinga, B. andMolendijk, M. (2020),“Anxiety, depression and post-traumatic stress disorder in refugees resettling in high-income countries: systematic review and meta-analysis”, BJ Psych Open, Vol. 6 No. 4, p. e68, doi: 10.1192/bjo.2020.5.
Blackmore, R., Boyle, J.A., Fazel, M., Ranasinha, S., Gray, K.M., Fitzgerald, G., et al. (2020), “The prevalence of mental illness in refugees and asylum seekers: a systematic review and Meta-analysis”,PLoS Medicine, Vol. 17 No. 9, p. e1003337, doi: 10.1371/journal.pmed.1003337.
Bogic, M., Njoku, A. and Priebe, S. (2015), “Long-term mental health of war-refugees: a systematic literature review”, BMC International Health and Human Rights, Vol. 15 No. 1, p. 29, doi: 10.1186/s12914-015-0064-9.
Gleeson, C., Frost, R., Sherwood, L., Shevlin, M., Hyland, P. and Halpin, R. (2020), “Post-migration factors and mental health outcomes in asylum-seeking and refugee populations: a systematic review”, European Journal of Psychotraumatology, Vol. 11 No. 1, p. 1793567, doi: 10.1080/20008198.2020.1793567.
Material & methods
The authors report on a qualitative study, however, there is a strong sense of quantitative methods in various parts of the paper. I will highlight as I go through
Again, I feel that there are too many sub-headings in this section
Study design heading describes using semi-structured interviews, can the authors begin by providing a brief overview of the actual methodology informing this. Also, it seems to be a purposive sample, with a Gatekeeper used, can these factors be expanded on, before moving onto sample etc, the semi structured interviews can be described later. The authors describe using Acculturation Theory to inform the interview schedule, is this an inductive or deductive study?
Data collection
What criteria were used to select translators? Were they trained translators or people from the native community who speak the language?
Measures
The authors collected a fairly large amount of survey data, why was this done for a qualitative study, what was the added value? I don't have an issue as such with collecting such information, however in the context of some other points that I will make, the study is verging on mixed methods
Data analysis
The authors don't provide information on the analysis the survey data, yet report on it in the results section? Again, leading me to believe the methodology needs reworking
Can the authors provide a couple of sentences illustrating the TA process from initial codes to themes.
Results
The authors provide mean and standard deviations on the sample characteristics, this is a quantitative method, and was not discussed in the data analysis section, and the study is presented as a qualitative study. The authors may wish to consider the added value of this with regard to a qualitative study
Qualitative results
Again, I have no issue with presenting percentage of respondents reporting on themes etc, but with the wider issues mentioned above, the mixed method framework becomes stronger
Themes 1,2 & 3
Can these themes be expanded on; there is not much data presented, and the richness of qualitative findings is a little absent. It seems that the authors have used the ages/demographics of participants in the theme section, to describe characteristics of participants during their analysis. Again, this is in keeping with a mixed methods study
Can authors italic direct quotes
Discussion
Overall, a good discussion. However, the literature review mentions mental health, and the discussion briefly mentions trauma is there any findings that need to be discussed here?
There should be more on practical recommendations for practitioners/organizations perhaps a heading on implications for practice, policy and research
I would also suggest being more explicit about how acculturation theory can support this practice/policy
I wonder if there is an avenue to highlight the role of cultural competency here. The study below may be helpful
Lau, L.S. and Rodgers, G. (2021), “Cultural competence in refugee service settings: a scoping review”,Health Equity, Vol. 5 No. 1, pp. 124-134, doi: 10.1089/heq.2020.0094.
Limitations
Qualitative research does not seek to be generalizable
Reviewer 2 Report
The article is interesting as it sheds some light on specificities of the acculturation process of congolese women after resettling in Ohio.
The introduction is well written, but it needs to include reference to the "place attachment" psychological construct - of paramount relevance here, especially when considering it in the light of identity (see different recent research articles, and particularly a recent study on an italian refugee sample which bears noticeable methodological similarities with the present study).
In the Discussion as well as in the Limitations sections, authors should address more clearly the difficulty to generalize their results - I am referring especially to the great variability in partecipants' age and year of arrival in the US .
Round 2
Reviewer 1 Report
The authors substantively addressed the feedback